# Behavioural Patterns and Postnatal Development in Pups of the Asian Parti-Coloured Bat, *Vespertilio sinensis*

**DOI:** 10.3390/ani10081325

**Published:** 2020-07-31

**Authors:** Deyi Sun, Yu Li, Zhongwei Yin, Kangkang Zhang, Heng Liu, Ying Liu, Jiang Feng

**Affiliations:** 1Jilin Provincial Key Laboratory of Animal Resource Conservation and Utilization, Northeast Normal University, Changchun 130024, China; sundy170@nenu.edu.cn (D.S.); liy476@nenu.edu.cn (Y.L.); yinzw123@126.com (Z.Y.); zhangkk307@nenu.edu.cn (K.Z.); liuh548@nenu.edu.cn (H.L.); 2College of Life Science, Jilin Agricultural University, Changchun 130118, China

**Keywords:** behavioural development, Asian parti-coloured bat, postnatal development

## Abstract

**Simple Summary:**

During the development of animals from juvenile to adult, their behaviour, morphology, and physiology can be altered to accommodate specific developmental periods. Early development in young animals directly affects the acquisition and maturation of behavioural skills and neurological functions in adults. Studying the behavioural development of young animals can help reveal the formation and evolution of animal behaviour. As the only mammalian group that flies, studying the behavioural development of young bats can help improve our understanding of bat flight behaviour. From our observations in the laboratory, we found that wing flapping and wing spreading behaviours in young bats promote earlier flight attempts.

**Abstract:**

Behavioural development is an important aspect of research on animal behaviour. In bats, many studies have been conducted on the development of flight behaviour, but the postnatal behavioural development of bats remains largely unexplored. We studied the behaviours and postnatal development of infant bats by conducting controlled video recorded experiments. Our results showed that before weaning, Asian parti-coloured bats (*Vespertilio sinensis*) were able to exhibit four types of behaviours, namely, crawling, head moving, wing flapping, and wing spreading, and these behaviours are different from those observed in experiments with adult bats. The number of occurrences of these behaviours was correlated with age and scaled mass index. Furthermore, the number of occurrences of these behaviours in young bats could also reflect their physical developmental status. In young bats, wing flapping and spreading might be a type of play behaviour. These behaviours were negatively correlated with the time of the first flight, indicating that they might help to promote individual physical development. Our results provide fundamental data for revealing the ontogenetic and neurophysiological mechanisms of behavioural development in bats.

## 1. Introduction 

Tinbergen [1] established behavioural development as one of the four main problems in behavioural biology. In mammals, adult behavioural patterns are often expressed by juveniles throughout ontogeny. For example, the songs of birds are formed gradually during their development [2]. Social behaviour during juvenile development in bottlenose dolphins (*Tursiops* spp.) affects their social relationships in adulthood. The behavioural repertoire of a mature animal cannot be fully understood without knowledge of the ontogenetic origins of the animal’s behaviour [3,4].

Bats live on all the continents except Antarctica and comprise one in every five mammalian species today [5,6]. The great diversity and extraordinary distribution of bats are mostly attributed to their wings and ability to fly [7]. Flight has enabled bats to exploit a variety of foraging niches inaccessible to other mammals [8,9,10,11]. Nevertheless, the development of flight remains one of the most remarkable and characteristic aspects of bat development [12]. Most echolocating bats are altricial, and after birth, their young undergo a period of growth whereby they develop the appropriate sensory and locomotor skills before becoming independent of maternal care [13]. There is little doubt that this postnatal growth period is a key growth stage for bats, because the growth and development of flight, wing structures, and echolocation calls occur mainly during this time [14]. Newborn infants develop quickly and thus have a short period to develop the flight skills necessary to successfully forage on nocturnal invertebrates. Weaning typically occurs within the first 3 weeks after birth, and juveniles must achieve sufficient flight abilities quickly thereafter or perish due to starvation [15]. 

There are many in-depth studies on the flight development of bats, such as studies on the morphological development of bat wings, forearm length, wingspan, wing area, and other wing characteristics [16,17,18,19,20,21,22,23,24,25]. Patterns of postnatal development in wing morphology, such as wingspan and wing area, usually increase linearly until the age of the first flight, when growth rates begin to decrease. Changes in body mass may dramatically affect flight capability [16,17,18,19,20,21,22,23,24,25]. Increases in body mass and wing loading result in decreases in flight speed and manoeuvrability [9,26,27] and an increase in energy costs [28]. Morphological development and body mass increase directly affect the development of behaviour. However, there is a lack of research on the behavioural development of bats. In mammals, some social behaviours develop before offspring are weaned. Strauss [29] studied the formation of courtship behaviour during the development of young greater white-lined bats (*Saccopteryx bilineata*) and found that they can grasp courtship behaviour by observing the behaviour of adult bats during development. However, juvenile behavioural development and its relationship with physical development and the acquisition of adult flight behaviour are largely unexplored.

Before a bat actually flies, it will perform push-ups or display stretch-and-push behaviour [30]. Practice or training before flight is important. Because simulated flight is unlikely to occur naturally during development because of the lack of space in most roosts, flight training appears to consist of abundant fluttering and stretching as practice flights [30]. Such types of pre-flight stretching or exercise have been observed in *Myotis velijier* [31], *Rhinolophus cornutus* [32,33], *Nyctalus noctula*, *Pipistrellus pipistrellus*, *Eptesicus serotinus* [34], and *Myotis lucifugus* [30,35]. Pre-flight training is very likely to be an integral part of flight ontogeny and important in aiding muscular development [35] and encouraging the ossification of wing bones [36]. Therefore, exploring the patterns and changing trends of behaviours during the postnatal development of young bats from birth to their first flight can help improve the understanding of how behavioural development affects the flight ability of bats.

The Asian parti-coloured bat (*Vespertilio sinensis*) is distributed in Korea, Japan, Russia, and China [6]. These bats are a highly complex community and often roost in clusters in holes along bridges or in the roofs and eaves of old buildings, preferring narrow spaces to wide caves [37]. Their main activity in the roost is crawling. Asian parti-coloured bats give birth in late June, and observations in the wild have shown that newborn bats can fly as early as 22 days of age [37]. Their small body size (15.99 ± 3.19 g) and rapid growth make the Asian parti-coloured bat an ideal species for research on behavioural development.

Therefore, the Asian parti-coloured bat was selected as the focal species in our study. By observing and identifying the behavioural patterns of the pups after birth as well as analysing the relationships between the number of occurrences of the behaviours and age, body mass index, and the acquisition of flight ability, the development of pre-flight behaviours and their influence on the flight ability of bats were examined. We predicted that: (1) the postnatal development of the behaviours of Asian parti-coloured bat pups was related to their physical condition and age, and (2) the behaviours of the pups before flight affected the acquisition of flight thereafter.

## 2. Materials and Methods

### 2.1. Sample Collection and Domestication

Previous observations showed that Asian parti-coloured bats begin to give birth in late June, and field observations have shown that after birth, young bats can fly clumsily as early as 22 days of age [37].

On 15 June 2019, under the Acheng overpass in Harbin, six female bats late in pregnancy (determined by palpating the abdomen; Kunz and Fenton, 2003) were captured by using mist nets when bats were out hunting at night. The captured females were brought to an indoor laboratory for feeding. The six female bats were divided into two groups (Group 1 and Group 2), and each group was acclimated to an enclosure with two connected cages (each cage had a size of 40 × 50 × 60 cm) (Figure 1). In each enclosure, one cage (the roosting cage) was covered with damp towels to create a hospitable environment. In the other cage (the foraging cage), food and water dishes provided mealworms and water from which the bats could feed. The two cages were connected through a 10 cm long channel. During the foraging process, the females usually left their infants in the roosting cage and foraged alone in the foraging cage. The design of the cage allowed the young bats the opportunity to freely display the necessary behavioural expressions. The females in Group 1 produced five pups (three male bats and two female bats), and the females in Group 2 produced six pups (five male bats and one female bat). Each pup was tagged with a ring (4.2 mm internal width, 5.5 mm height; Porzana Ltd., Winchelsea, UK) to distinguish the individuals. The ontogenesis of individual behaviours was recorded and analysed from all the pups. After the experiment was completed, the pups and their mothers were released at the sample site together. The capture of bats was performed under license from the local government.

### 2.2. Behaviour Observation and Recording

Pups are very vulnerable at birth; therefore, the study was conducted with pups beginning at three days old to avoid causing mortality due to improper manipulation. To ensure that each bat was able to fly independently, the experiment was conducted until the end of 35 days. Video collection was carried out from 8:00 pm to 8:00 am the next morning every day. Infrared cameras (Sony Digital Handycam HDR-CX760E) were used to record videos of the bat behaviours. Two observers (Sun and Li) independently observed the video and judged the types of behaviours displayed by the infants. The video was played at normal speed (50 frames per second). The observers watched the video carefully and recorded the behaviours of each individual infant after identifying them by their ring marker (individuals are distinguished by the position and number of ring markers). The number of occurrences of these behaviours was calculated for each individual on each day. Crawling was recorded when an individual infant crawled away from the group and returned quickly without a purpose. Head moving was recorded when one infant began to shake his or her head within a small range. Wing spreading was recorded when an infant opened one or both wing membranes without flapping. Wing flapping was recorded when an infant flapped its wings continuously within a given period (Appendix A) has more detailed video information). In all the behavioural videos recorded, the individual bats mainly show antagonistic interactions, and there are no other obvious communication behaviours. Therefore, no inter-individual communication behaviour was recorded in this experiment.

### 2.3. Scaled Mass Index

During the development of young bats, their physical condition also changes. To test whether the physical condition of young bats can affect behavioural development, we regularly checked the physical parameters of the young bats and calculated the body mass index. In bat ontogeny studies, body weight and forearm length are usually used as parameters to monitor growth and development [37]. Once they were three days old, the body weight and forearm length of the infants were measured with an electronic balance (Ohaus LS 200, 0.01 g) and electronic Vernier calipers (Tesa Cal IP67, Switzerland, 0.01 mm) every 2 days to monitor their development. During development, body weight and forearm length change significantly with age, and calculations using the traditional body mass index formula are influenced by the effects of body development on body mass index. To avoid the deviations caused by body development itself, a scaled mass index was used to calculate the body mass indexes at different ages. The formula of the scaled mass index is as follows: ${\hat{M}}_{i}=M_{i}{\left[\frac{L_{0}}{L_{i}}\right]}^{b_{SMA}}$, where ${\hat{M}}_{i}$ is the scaled mass index, $M_{i}$ is the weight of infants, $L_{i}$ is the forearm length of infants, $L_{0}$ is the average of the infants’ forearm lengths, and $b_{SMA}$ is a proportional [38] coefficient.

### 2.4. Flight Ability Experiment

Flying is a very challenging skill, and it has strict requirements for body structure and motor coordination [39]. In addition, two important benchmarks for successful postnatal growth and development in bats are the attainment of flight and independence from parental care [14]. It was observed that the movement patterns of wing flapping and wing spreading in infant Asian parti-coloured bats are the same as the flight patterns in adult bats. To test whether the flapping and spreading behaviours of young bats during development are related to the bat’s first attempt at flight and its ability to successfully fly independently, we designed a flight ability experiment. A previous study has shown that young bats in the wild exhibit flight behaviour at the age of 22 days [37]. According to the developmental patterns (body weight and forearm length) of the bats used in this study, the experiment was conducted before the infants reached 20 days of age. The flight experiment began when the young bats were 17 days old. Mimicking their habitat in the wild, a platform was designed to allow the young bats to hang upside down and was lined with cushioning foam to prevent injuries if they fell (Figure 2). At the start of the experiment, young bats were placed on the take-off platform by researchers and then their behaviours were observed. The young bats would either fly off or stay on the take-off platform. The researchers monitored the positions of the young bats to determine if their behaviour counted as an observation (staying on the take-off platform for more than three minutes counted as an observation). Each young bat was tested for flight ability three times a day before the daily behavioural observations began.

The full behavioural assessment of the young bats was carried out according to the process described in Figure 3. The experiment ended when all the individuals achieved successful flight and a successful landing. The flight performances of the young bats were monitored according to the criteria in Table 1, and the ages of the young bats as they completed each stage were recorded.

### 2.5. Statistical Analysis

A generalized linear model was used to examine whether the behavioural development of young bats was influenced by age, scaled mass index, or the interaction between age and scaled mass index (using observation-level random effects to reduce overdispersion errors) [41]. Correlation analysis was used to examine the relationship between the flight ability of young bats and the number of behavioural occurrences. The data were tested for normality before analysis (Shapiro–Wilks test, α = 0.05). Pearson correlation analysis was used to analyse the two sets of data that conformed to the assumptions of normality (Pearson correlation coefficient). Spearman correlation analysis was used for the two sets of data that did not conform to the assumptions of normality (Spearman correlation coefficient).

To exclude the interaction between age and scaled mass index, partial correlation analysis was conducted between the two variables and the number of occurrences of the behaviours, controlling for age and scaled mass index separately (95% confidence interval). Polynomial analysis was used to fit the curve equation, and the age of young bats was used as the independent variable to observe the change in the number of occurrences of wing flapping and wing spreading behaviours with age. The abovementioned statistical analyses were performed in R (R version 3.6.0), Matlab (Matlab version 2013b), and SPSS (SPSS version 22, SPSS Inc., Chicago, IL, USA). The scaled mass index of the bats was calculated every other day during the developmental process. In the analysis involving the scaled mass index, the number of occurrences of the behaviours was taken from the day on which the scaled mass index was calculated.

## 3. Results

The behaviours of the 11 infants were recorded for 33 days, and 420 h of video was obtained. Integrating the results recorded by the two observers, four kinds of locomotor behaviours, i.e., crawling, head moving, wing flapping, and wing spreading, were classified in Asian parti-coloured bat infants in the early stage of development (3–35 days old) (Table 2).

As the infants developed, the number of occurrences of the behaviours also changed daily (Appendix A). As shown in Figure 4, according to similar changing trends and combined with the behavioural patterns of the young bats themselves, the four behaviours are divided into two categories. Crawling and head moving were classified into one category, while wing flapping and spreading were classified into another category. Crawling and head moving both reached their maximum number of occurrences at the age of 6 days and then showed a decreasing trend. After the age of 17 days, the young bats no longer exhibit crawling or head moving behaviours. The wing spreading and flapping behaviours increased rapidly with age, reaching a peak at approximately 15 days, after which the occurrence of these behaviours decreased until stopping completely after 25 days (Figure 4).

The number of occurrences of the crawling and head moving behaviours was influenced by the scaled mass index (*Z* = 5.62, *p* < 0.001), age (*Z* = −3.62, *p* < 0.001), and the interaction between scaled mass index and age (*Z* = −3.62, *p <* 0.001). When the age of the infants was controlled, there was a significant positive correlation between the scaled mass index and crawling and head moving behaviours (*R =* 0.34, *p* <0.001). Furthermore, when the scaled mass index was controlled, there was a significant negative correlation between the age of the infants and crawling and head moving behaviours (*R =* −0.39, *p* < 0.001) (Figure 5).

The number of occurrences of the wing spreading and flapping behaviours was influenced by the scaled mass index (*Z* = 4.16, *p* < 0.001), age (*Z* = 4.27, *p* < 0.001), and the interaction between scaled mass index and age (*Z* = 4.27, *p* < 0.001). Due to the significant interaction between age and scaled mass index, a partial correlation analysis was performed to control for the interaction between these two factors. When the age of the infants was controlled, there was a significant positive correlation between the scaled mass index and the wing spreading and flapping behaviours (*R =* 0.47, *p* < 0.001). Furthermore, when the scaled mass index was controlled, there was a significant positive correlation between the age of the infants and the wing spreading and flapping behaviours (*R =* 0.19, *p =* 0.03) (Figure 6).

The results of the polynomial analysis showed that the fitted curve equation for the number of wing spreading and flapping behaviours with age was: f(x) = −0.3341x^2^ + 9.706x − 30.34, (x: age, *R^2^* = 0.75, Adj *R^2^* = 0.74). The number of occurrences of the wing spreading and flapping behaviours in young bats increases and then decreases with age (Figure 7). 

The correlation analysis showed that there was a significant negative correlation between the frequency of the wing spreading and flapping behaviours and the age of the individuals at their first flights (Spearman’s rank correlation, *R* = −0.46, *p* = 0.01). The young bats that exhibited more wing spreading and flapping behaviours tried to fly earlier. However, there was no significant correlation between the frequency of the wing spreading and flapping behaviours and the time it took for the young bats to acquire flight ability (Spearman’s rank correlation, *R =* 0.25, *p =* 0.46). The young bats that exhibited more wing spreading and flapping behaviours did not necessarily gain the ability to fly freely faster (Figure 8).

## 4. Discussion

Throughout the behavioural observation period, four kinds of locomotor behaviours were found in the young Asian parti-coloured bats: crawling, head moving, wing flapping, and wing spreading. The frequency of these four behaviours also changes as the young bats develop, and their mothers rarely exhibit them. Among these four behaviours, crawling and head moving were mainly observed in newborn bats. The infants crawled for short distances and then retreated to their starting positions, showing no obvious purpose, while adult bats usually crawl towards food or other targets. Infants shake their upper bodies vigorously when they move their heads. The head movements of adult bats are usually confined to their heads, and their upper bodies do not move. The wing flapping and spreading behaviours of the young bats are more obvious than the other behaviours. With an increase in the age of the infants, the wing flapping and spreading behaviours first increased and then decreased. These activities involve the patagium. 

All four behaviours were observed to disappear later in development, rather than increase as the infants matured. Therefore, we speculate that these behaviours are developmental behaviours of young bats, and they will gradually disappear as the bats mature. Oppenheim [42] suggested that developmental stages are often not merely a kind of immature preparation for the adult state—although they certainly can be—but that each developmental phase involves unique adaptations to the environment of the developing animal. Oppenheim [42] called these “ontogenetic adaptations”. As a consequence, certain early behavioural patterns may disappear in the course of development, a phenomenon that Oppenheim [42] termed a “retrogressive process”. The change in the trends of the infant behaviours in the experiment may have been a retrogressive process.

The example of imprinting in birds reveals the kinds of interactions that can take place between internal processes and external conditions during the course of development [43]. The results of the generalized linear model showed that all four behaviours of infant bats were influenced by age, scaled mass index, and the interaction between them. A partial correlation analysis further revealed the relationship between the various behaviours and influencing factors. With increasing age, the number of occurrences of crawling and head moving in the young bats gradually decreased, and there was a significant negative correlation between these factors, especially in the early stage of young bat development. Although the partial correlation analysis showed that wing flapping and spreading were positively correlated with the age of young bats, combined with Figure 7, it was determined that the wing flapping and spreading behaviours of the young bats first increased and then decreased. This suggested that with the increase in the ages of the young bats, especially in the middle and late stages of development, the behaviours observed were mainly wing flapping and spreading. However, the different types of behaviour might be related rather than isolated from each other. 

Animal development is seen as a complex series of interactions between an individual and its internal and external environments; developmental changes at any time are affected by the preceding developmental stage, which itself is the product of a complex interaction between the individual at a certain time and its environment [44]. In young bats, the occurrences of different types of behaviours reach their peaks in different periods, which may be related to morphological development and neuromuscular maturity [37]. One kind of behavioural process may have to precede another if the second kind is to function properly [43]. The wing flapping and spreading behaviours are more complex than the crawling and head moving head behaviours. Therefore, in the process of bat development, it is necessary to first develop simple crawling and head moving behaviours before more complex behaviours, such as wing flapping and spreading. For example, in the little brown bat (*Myotis lucifugus*), animals do not spontaneously flap their wings during freefall until approximately 10 days of age, suggesting that the neuromuscular mechanism of the flap reflex is not developed until then [35]. Similarly, in-lab studies in rats have shown that the end of the postnatal transition is the most critical period of development, when all aspects of neural maturation suddenly reach adultlike perfection [45].

In addition, all four behaviours were positively correlated with the scaled mass index, indicating that healthy young bats were able to develop more behaviours than unhealthy bats. Individuals can perform more behaviours when they are well nourished and energetic [46]. The frequency of animal behaviour itself is one of the indicators of animal welfare, and activity levels reflect the physical conditions of young bats [47]. The reflex test in mice can be useful in assessing the degree of neural maturation in development and are reliable indicators of normal development [45]. Therefore, the number of occurrences of behaviours in the Asian parti-coloured bat pups can possibly be used as an indicator of their health and developmental status. 

The behavioural development of young bats does affect their flight ability. The results showed that there was a significant negative correlation between the age of the first flight and the proportion of wing flapping and spreading behaviours. However, there was no significant correlation between the proportion of wing flapping and spreading behaviours and the time it took for the bats to acquire flight abilities. This suggests that Asian parti-coloured bat pups that engage in more wing flapping and spreading behaviours during development attempt to fly earlier, but it does not mean that flight behaviour develops faster. For bats, the time it takes them achieve their first flight represents the tendency to move, reflecting how active the young bats are in flight. Young bats that exhibit more wing flapping and spreading make earlier flight attempts, thereby achieving their first flight earlier than bats that exhibit less of these behaviours. The duration of successful flights reflects a more advanced flight ability in bats [12]. Flying is a very difficult skill to master, and it requires a high degree of body development and movement coordination [39]. In avian studies, it has been found that the modular movement strategy, which involves a trade-off between wing and leg movements, is more beneficial to the flight of birds [48]. Bat research has also found that the tail membrane plays an important role in flight, providing thrust and perhaps lift during flight [49]. This is reflected in the time it takes for young bats to achieve successful flight.

Many previous studies have used the Powers [35] method for testing the flight ability of bats [12,14,25]. This method focuses on the development of nerves and muscles in combination with wing morphology and movement in young bats, but it does not reflect the reality of young bats going from non-flying to being able to fly. Our experiment referred to Bell’s [40] study on fruit bats (*Pteropus* spp.). This method gives a more realistic picture of how young bats acquire the ability to fly. The results obtained by this method better reflect the relationship between juvenile bats’ behavioural development and flight behaviour.

The crawling and head moving behaviours of young bats may be an adaptation to and exploration of the environment and disappear when young bats are familiar with their living environments. Wing flapping and spreading behaviours, which gradually cease with the development of young bats after their first flights, are related to wing membrane movement. The development of motor skills is related to the maturation of the nervous and muscular systems [35]. Bats have very small wing areas and wingspans at birth, which do not meet the great power requirements of flight [32,35,50]. In this experiment, the proportion of wing flapping and spreading behaviours was not high in newborn bats; however, as the bats increased in age, the proportion of wing flapping and spreading behaviours increased rapidly and reached its peak at approximately 15 days of age. By this time, the young bats had not yet begun to fly. This type of behaviour, which occurs outside of a clear functional context, may be play behaviour [51,52]. Play is an interesting example of “behaviour in development”. It tends to occur at a particular stage in life but, as a result of the information and skills picked up at that time, almost certainly has a long-term influence on adult behaviour [43]. Although the body is not yet functionally developed enough for flight, we propose that wing flapping and spreading behaviours during development, similar to play behaviour, may prepare young bats for flight.

## 5. Conclusions

Our study found that young Asian parti-coloured bats exhibited crawling, head moving, wing flapping, and wing spreading behaviours. The behavioural development of young bats was affected by age and scaled mass index. Young bats that exhibit more wing flapping and wing spreading behaviours tend to fly earlier. The wing flapping and wing spreading behaviours might be a kind of play behaviour in young bats. Since indoor domestication of young bats is still difficult, this study was carried out only with Asian parti-coloured bats, which are relatively easy to domesticate. Future research could consider studying the development of young bats in more bat species, and observe the development of young bat populations under natural conditions to obtain richer data on the development of young bats. These studies can shed more light on the adaptive significance of juvenile bat behaviour and development and can lead to targeted bat conservation measures.

## Figures and Tables

**Figure 1 animals-10-01325-f001:**
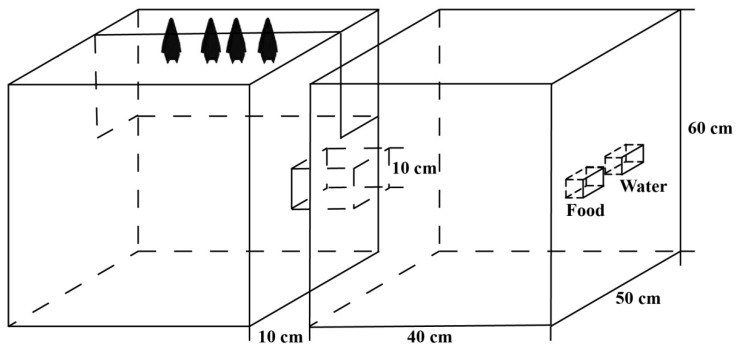
The two connected cages. One cage (the roosting cage), which served as a living environment, was covered with damp towels, while the other cage (the foraging cage) served as a feeding environment. Food and water dishes were used to provide mealworms and water. The two cages were connected by a channel with a length of 10 cm. The bat images in the figure represent only the position of the bats, not the exact number of individuals.

**Figure 2 animals-10-01325-f002:**
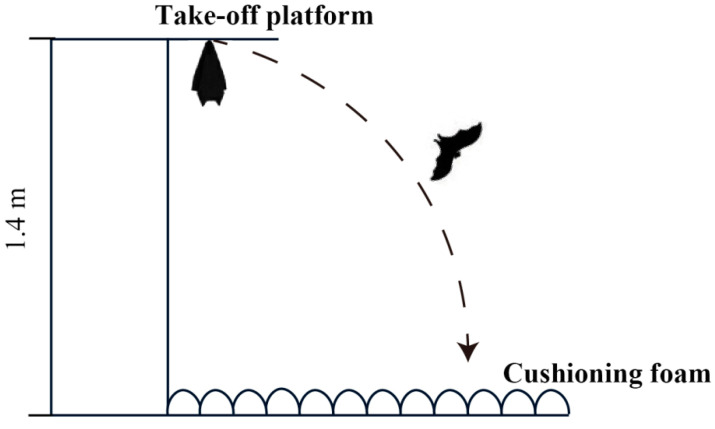
Schematic diagram of the flight experiment equipment.

**Figure 3 animals-10-01325-f003:**
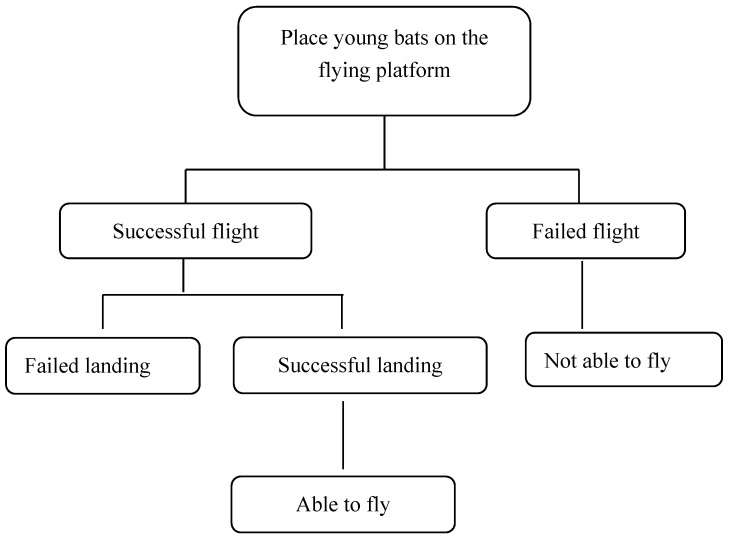
Flight capability determination process.

**Figure 4 animals-10-01325-f004:**
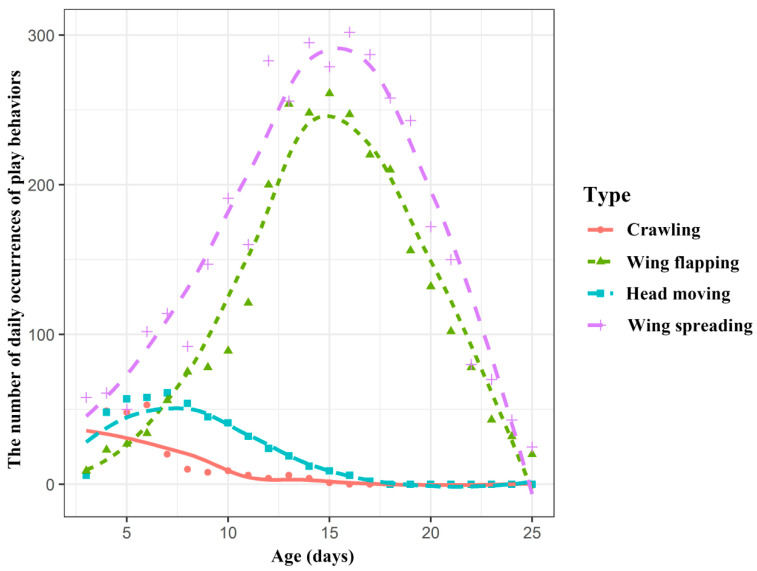
The number of occurrences of the different types of behaviours changed throughout the early development of the Asian parti-coloured bat (*Vespertilio sinensis*). The x-axis is the age of the infants, and the y-axis is the number of occurrences of the different behaviours in all the infants at a given age. The different lines represent the different types of behaviours.

**Figure 5 animals-10-01325-f005:**
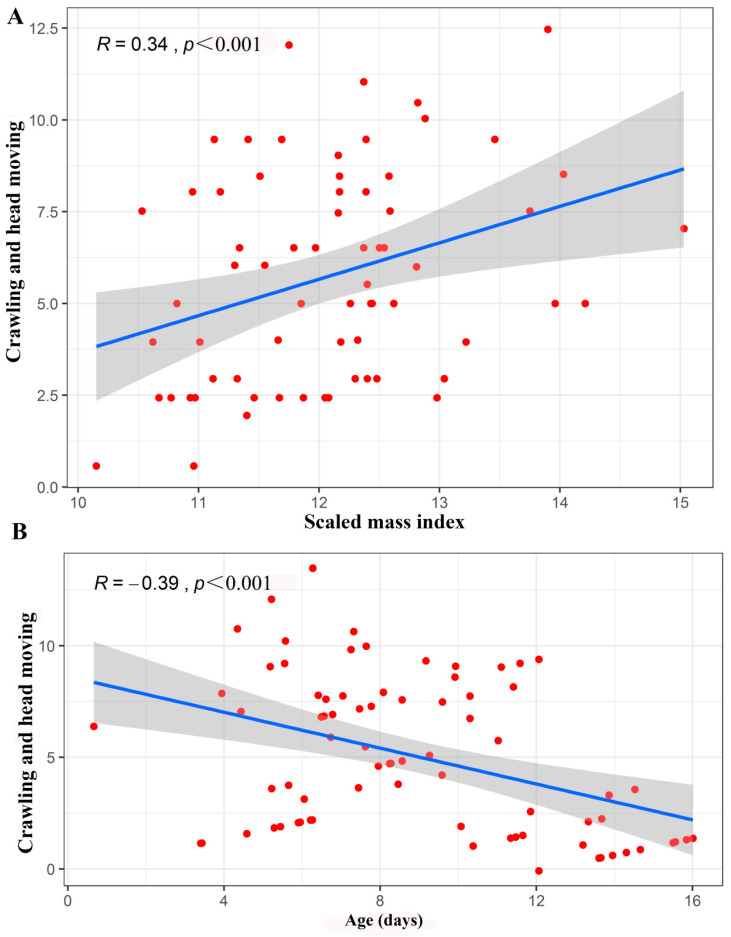
**(A**) Scatter plot of the partial correlation between the scaled mass index and number of occurrences of the crawling and head moving behaviours in young Asian parti-coloured bats (*Vespertilio sinensis*). (**B**) Scatter plot of the partial correlation between age and number of occurrences of the crawling and head moving behaviours in young Asian parti-coloured bats (*Vespertilio sinensis*). The solid blue lines represent the fitted curves, and the shadowed areas represent the 95% confidence interval.

**Figure 6 animals-10-01325-f006:**
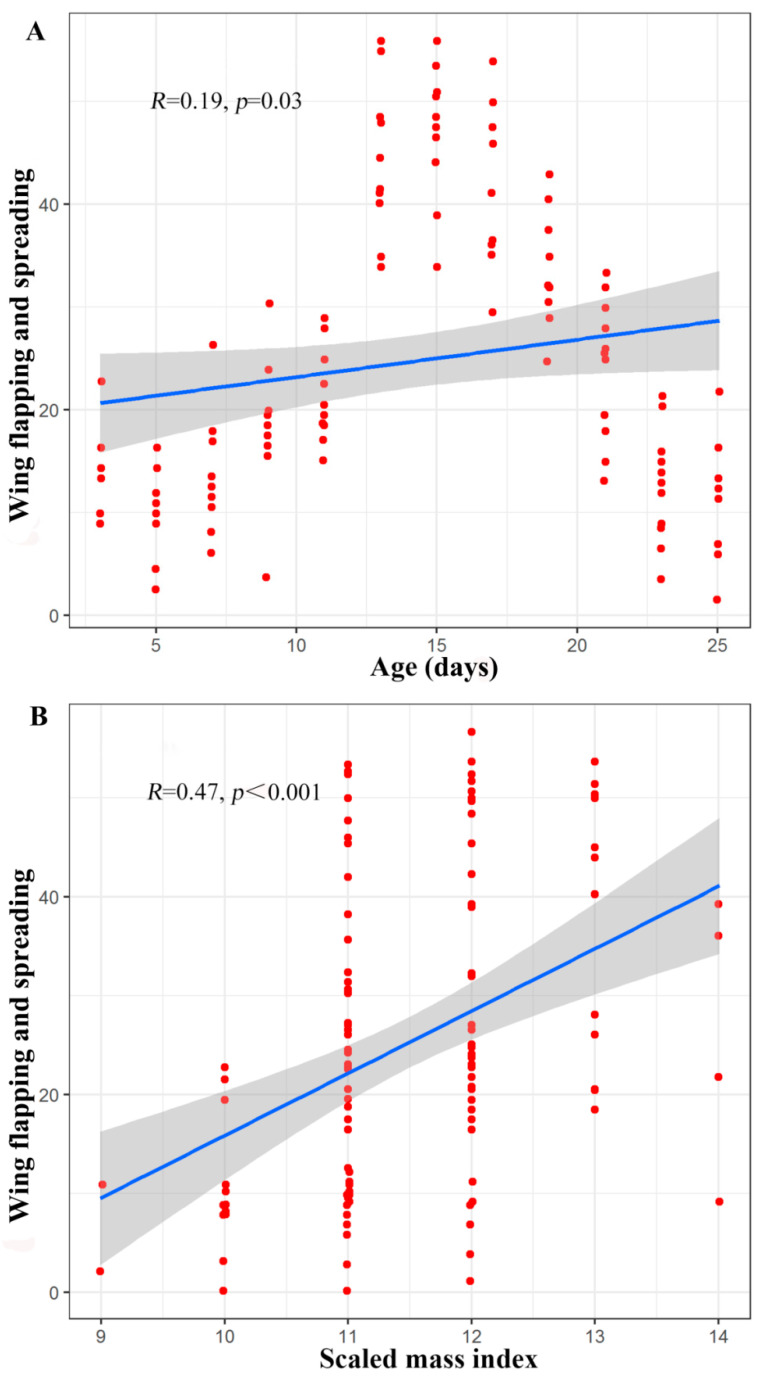
(**A**) Scatter plot of the partial correlation between age and number of occurrences of the wing spreading and flapping behaviours in young Asian parti-coloured bats (*Vespertilio sinensis*). (**B**) Scatter plot of the partial correlation between the scaled mass index and number of occurrences of the wing spreading and flapping behaviours in young Asian parti-coloured bats (*Vespertilio sinensis*). The solid blue lines represent the fitted curves, and the shadowed areas represent the 95% confidence interval. The flight experiments began when the bats were 17 days old, and there was a sequence of ages at which the individuals made their first flights (first successful flight, first failed landing). The time between the first flight and the acquisition of flight capability (successful flight, successful landing) varies between individuals (Appendix A).

**Figure 7 animals-10-01325-f007:**
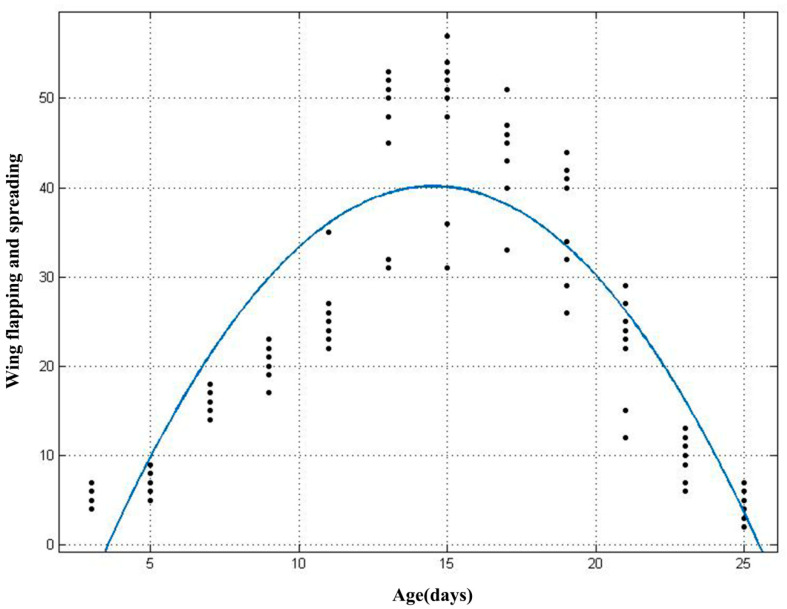
Fitting graph of age and the number of occurrences of wing spreading and flapping behaviours in young Asian parti-coloured bats (*Vespertilio sinensis*).

**Figure 8 animals-10-01325-f008:**
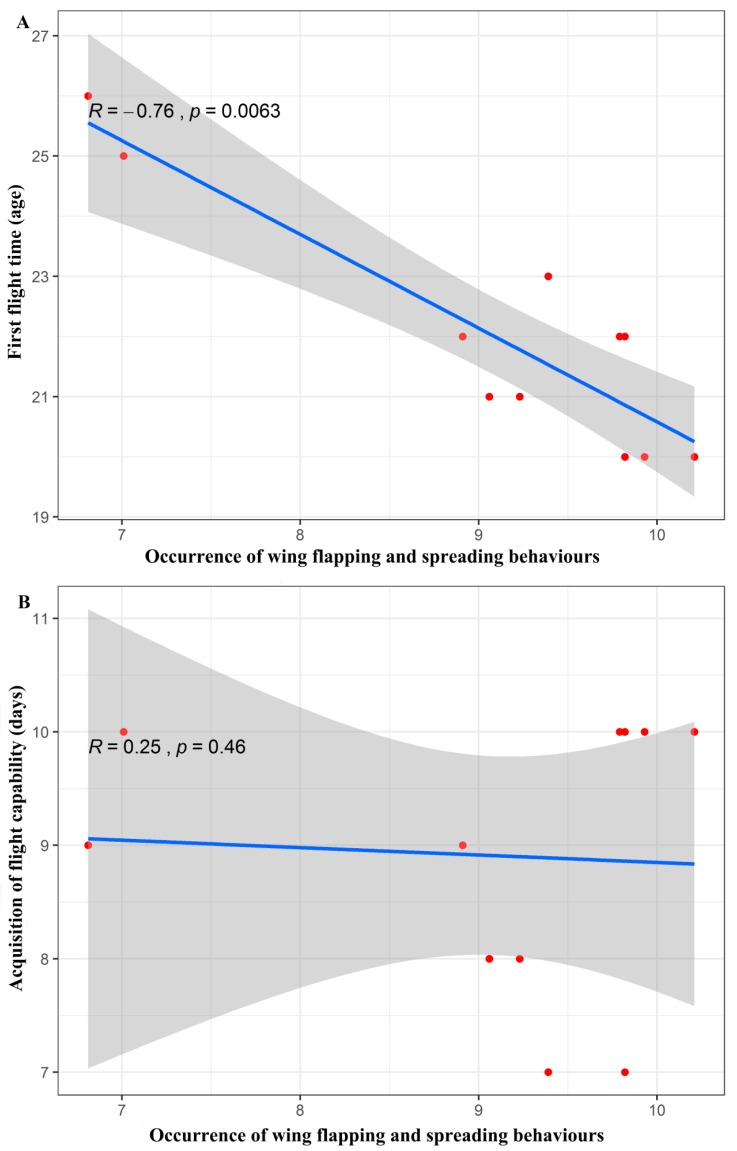
(**A**) Scatter plot of the correlation between the occurrence of the wing flapping and spreading behaviours and the age at which the young Asian parti-coloured bats (*Vespertilio sinensis*) performed their first flights. (**B**) Scatter plot of the correlation between the occurrence of the wing flapping and spreading behaviours and the time at which the ability to fly was acquired. The solid blue lines represent the fitted curves, and the shadowed areas represent the 95% confidence interval.

**Table 1 animals-10-01325-t001:** Judgement criteria for the flight conditions of young bats [40].

Behaviour	Standard	Description
Flight	Success	Young bats take the initiative to leave the flight platform and fly smoothly through the air.
	Failure	Young bats stay on the flight platform or fall off it uncontrolled.
Landing	Success	Young bats stabilize their limbs and body immediately before landing in a smooth and confident movement.
	Failure	Young bats land uncontrolled, usually accompanied by extending their patagium or making an additional flapping motion.

**Table 2 animals-10-01325-t002:** Description of the behaviours (ethogram) recorded in Asian parti-coloured bat infants.

Type	Number of Occurrences	Percentage	Description
Wing flapping	2715	38%	Infants open one or both wings and then flap them at a high frequency. The flapping of the wings occurs continuously, with intervals between each flap. This is occasionally accompanied by head movements.
Wing spreading	3718	52%	Infants open one or both wings without flapping, sometimes opening and closing the wings and sometimes leaving them open for a period of time. When they spread their wings, the infants are usually not performing other behaviours.
Crawling	225	3%	Infants move away from the other individuals and crawl around the cage. This behaviour would happen when the female had left to find food. The infants did not stray very far from their starting point, and then they returned to the group.
Head moving	474	7%	Infants push their heads up and look around. The range of movement frequency during head moving play was much wider than that of adults when perceiving their surroundings.

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
