# Peer review of "Behavioural Patterns and Postnatal Development in Pups of the Asian Parti-Coloured Bat, Vespertilio sinensis"

_animals, 2020, doi:10.3390/ani10081325_

Round 1
Reviewer 1 Report
See comments on attached manuscript

Author Response
Response to Reviewer 1 Comments
Thanks for your constructive comments and valuable suggestions on our manuscript. We have carefully read the comments and suggestions, and have made detail corresponding revisions. Point-by-point responses are attached below.
Point 1: as the only mammal group
Response 1: Thank you very much for the suggestion. The statements in the article have been changed. Please see line 14.
Point 2: change from "hot topic" to "important aspect of animal behaviour research".
Response 2: Thank you very much for the suggestion. The statements in the article have been changed. Please see line 18.
Point 3: need Latin name on first use
Response 3: Thank you very much for the suggestion. The statements in the article have been changed. Please see line 23.
Point 4: Is this the common name?
Response 4: I'm sorry about that. It was an error in the writing process. The statements in the article have been changed. Please see line 70.
Point 5: need citation if possible
Response 5: Thank you very much for the suggestion. The statements in the article have been changed. Please see line 89.
Point 6: I would caveat this - doesn't really mimic a foraging environment.
Response 6: It's a very useful suggestion. These statements in the article may be inaccurate and ambiguous. I've changed these statements in the article. Please see line 114.
Point 7: Could you not also model this in a generalized linear model using an ordinal distribution?
Response 7: It's a very useful suggestion. Since the sample size in the experiment is still relatively small and we only want to examine the relationship between the two variables. So using correlation analysis may be a more appropriate option.
Point 8: I don`t know that you can say without a clear purpose. I would delete
Response 8: Thank you very much for the suggestion. The description of the behaviours is intended to show that these behaviours are unique to young bats. Even though adult bats exhibit similar behaviours, the extent of the behaviours and the context in which they occur are different from those of juvenile bats. These statements in the article may be inaccurate and ambiguous. I've changed these statements in the article. Please see lines 235-236.
Point 9: each table or figure that uses common name should also include Latin name so that legends can stand alone from text.
Response 9: Thank you very much for the suggestion. The statements in the article have been changed. Please see lines 247, 330, 317, 329, 343.
Point 10: suggests a quadratic or some other way to model would have been more appropriate.
Response 10: Thank you very much for the suggestion. This is a very critical suggestion. The scatter in the figure is closer to the quadratic distribution. I've added quadratic analysis to the methods and results section. Please see lines 222-225, 305-308.
Point 11: So again each table or figure legend needs to be able to stand alone from text, so information here is incomplete without study animal, etc.
Response 11: Thank you very much for the suggestion. The statements in the article have been changed. Please see lines 247, 330, 317, 329, 343.
Point 12: I may have missed this in the methods - did you record ultrasonic vocalizations?
Response 12: I'm sorry about that. It was an error in the writing process. In previous pre-experiments we used an ultrasonic recorder. The results of the pre-experiment showed that the sounds made by young bats were not well collected, so the ultrasonic recorder was not used in this experiment. The statements in the article have been changed. Please see lines 356-359.
Point 13: probably needs a citation?
Response 13: Thank you very much for the suggestion. The statements in the article have been changed. Please see line 392.
Point 14: be careful - this is one species, you might want to caveat a bit and say " For example, in the little brown bat (Myotis lucifugus)....."
Response 14: Thank you very much for the suggestion. The statements in the article have been changed. Please see line 395.
Point 15: I might say simply delete all the play stuff and just end with these behaviors are critical for practice/honing life skills, muscle development and so forth.
Response 15: Thank you very much for the suggestion. The statements in the article have been changed. Please see lines 449-461.
Reviewer 2 Report
Overview
The authors have provided a manuscript (MS) that details behavioural development of infant Asian parti-coloured bats. A total of 11 infant bats and their mothers (six adults) were held in a temporary enclosure and the frequency at which the infant bats displayed four different behaviours (crawling, head moving, wing flapping and wing spreading) were monitored passively using IR video cameras over 33 days. Once the infant bats were 17 days old, a further experiment was conducted to document progress in flight attempts and increasing flight success/ability. The findings showed that two of the observed behaviours of young bats (wing flapping and wing spreading) promoted earlier flight attempts. The authors propose that these behaviours may help to promote the physical development required by juvenile bats to attain flight ability shortly after they are weaned. This is an interesting study that makes a valuable contribution to the field of behavioural and physical development in bats. My only concern with the manuscript was that, in the Dicsussion, the authors directly compare behviours of infant bats with adult bats. However, the rationale for making these direct comparisons was not established in the Methods, and the data used to make these comparisons were not described in the Results.
Specific comments by section.
Simple Summary
No comments.
Abstract
Lines 22-24: If you are going to compare behaviours of juvenile and adult bats then you need to describe the adult behaviours in the Methods. And if you are directly comparing the infant and adult behaviours then you need to explain what data are being used to make these comparisons, either descriptively or empirically using statistical tests.
Introduction
Lines 37-38: Include citation number immediately after the author’s name:
“Tinbergen [1] established behavioural development as one of the four main problems in behavioural biology.”
Lines 53-55: I would suggest using the term ‘nocturnal invertebrates’ rather than “night-flying insects”. Some bat species glean non-flying prey from vegetation, including taxa that are not insects, e.g. spiders.
Lines 44-71: The second paragraph is very long. I would suggest splitting it into to shorter paragraphs. A logical place may be line 57:
“There are many in-depth studies on the flight development…”
Lines 59-61: This sentence probably need a reference.
Lines 72-73: This statement needs to be referenced.
Line 84: Looks like there’s been an autocorrect of the spp name here:
“Vespertilio sine12nsis”
Lines 89-91: Would be useful to include the mean adult body mass of the spp here.
Lines 96-99: In this sentence I would add colon before the first prediction, and then a comma before the second prediction.
“We predicted that: 1) the postnatal development of the behaviours of Asian parti-coloured bat pups was related to their physical condition and age, and 2) the behaviours of the pups before flight affected the acquisition of flight.”
Materials & Methods
Line 102: You don’t need to include the spp. name again here, as you have already provided it in line 84.
Lines 102-104: Do you have a study you can cite here? If not, probably worth including a reference to the author’s ‘unpublished data’.
Line 105: spell ‘six’ rather use the numeral (6).
Lines 107-108: Was the laboratory an indoor or outdoor facility?
Lines 115-117: So were there three female bats in Group 1, and three bats in Group 2? Do adult female Asian parti-coloured bats normally give birth to single young, or twins? Presumably they have twins, given there were 11 pups from 6 mothers? Do did one pup die, or did one of the mothers only have a single pup?
Lines 120-121: List the license/permit number that the work was conducted under here. What about approval from the relevant Animal Ethics Committee? Presumably both Northeast Normal University and Jilin Agricultural University require any research involving the capture and handling of wildlife animals to be approved by their Animal Ethics Committee?
Lines 133-135: I’d recommend including the initials of the two observers here in brackets.
Line 135-136: Were the ring markers individually colour-coded? I presume they would have too small for the observers to be able to read any sort of number/ID code?
Lines 164-166: Can you clarify whether this observation has been made during previous studies, or by the authors during this study?
Lines 168-169: Only one study is referenced, so I’d say that “A previous study…”, instead of “Previous studies…”.
Line 186: The citation at the end of the caption (Edward 2019) is formatted differently to the journal style guide.
Results
Lines 306-310: In the captions for figures 5, 6 & 7 insert a bracket after A), and then B).
Line 296: This text is in the Results, so I don’t think you need to start the sentence with “The results showed that…”.
Better to start this sentence with: “The number of occurrences of…”
Line 321: Again, just start this sentence with: “The correclation analysis showed that…”
Lines 323-324: Swap “try” with ‘tried to fly earlier’.
Discussion
A general comment – the first two paragraphs in the Discussion are very long. Id recommend breaking these up into shorter paragraphs.
Lines 338-340: When you say “adult bats”, are you referring to the frequency of behaviours recorded when the infant bats were older, i.e. later in the experiment (i.e. behaviours of infant bats recorded on day number 25)? I can’t se anywhere in the Methods or Results any mention of behaviours of the 11 infant bats being compared to the behaviours of the 6 adult mothers being recorded. None of your models compared behaviours of infant bats with behaviours of their adult mothers. If you’re going to formally compare infant and adult behaviours then this needs to be described in the Methods and Results. If you’re just talking about anecdotal observations that’s fine, but you need to explicit about this, and then you can’t say that infant behaviours were “significantly different” to adult behaviours.
Lines 343-344: Did you use an ultrasonic detector to check whether they emitted an ultrasonic vocalisations? If not, then need to say here that the infant bats did not make any sounds that were audible to the authors.
Lines 343-345: Again, did you use an ultrasonic detector to record ultrasonic volalisations made by adult bats? Or is this a hypothesized behaviour?
Lines 349-351: Again, here you’re talking about observations of adult bat behaviours, but you have not described anything about these observation s in the Methods or Results sections. Even if these observations were essentially anecdotal, if you’re going to use them in the Discussion to directly compare with the infant bevaviours that you describes and analysed then you need to also describe the adult bahviours in the Methods, and at least provide some sort of descriptive summary of the adult behaviours that you observed in the Results.
Lines 357 & 359: Include the citation number each time you refer to Oppenhiem [41].
Line 408: I’d be good to include citations to a couple of example studies that have used the Powers [35] method at the end of this sentence.
Lines 408-416: I found this paragraph a bit difficult to follow. I’d suggest a rewrite to make it a bit clearer.
Lie 248: Include the citation number when referring to Burghardt [51].
Lines 435-437: I think you probably need to add here that you are proposing this, i.e.:
Although the body is not yet functionally developed enough for flight, we propose that wing flapping and spreading behaviours during development, similar to play behaviour, may prepare young bats for flight.”
Lines 439-440: But you didn’t describe any adult behaviours in the Methods or Results, and didn’t compare the bevaviours of infants with those of adults. Also, I’m not sure that the term “serious behaviour” is appropriate? I can’t see any way that you could objectively quantify whether any specific behavior of an infant or adult bat was “serious” or not.
Lines 443-445: delete “, in order to protect bat populations” from the end of this sentence, it’s not clear how this relates?
Lines 445-450: This is a very long sentence that I would suggest might be better broken into two sentences.
Line 459: Better to say that “Kangkang Zhang and Heng Liu analysed the data”
Lines 451-452: These videos are very helpful in providing the reader with a visual depiction of the four different behaviuours. Reference should be made to these four videos in the Methods section when you first describe the four different behaviours – I think you want the reader to watch the videos before they go on to read the Results and Discussion sections.
References
Be consistent with the dash size used in page numbers. Some references use 1-10, others 1–10.
Line 492: Reference 12 – Jones is spelt inccorectly (Jonesg).
Line 497: Reference 14 – the year should be in bold.
Line 527: Reference 26. Unclear whether this is a book, a book chapter, or a journal paper?
References 31 & 33: Why are the titles of these two journals abbreviated, but other journal titles are presented in full?
Line 570: Reference 41 – All words in the Journal title should be chaptalized.
Lines 572-573: Reference 42 – What journal is this paper from?
Line 586: Reference 48 – Delete the full stop between the end of the article title and the journal name.
Author Response
Response to Reviewer 2 Comments
Thanks for your constructive comments and valuable suggestions on our manuscript. We have carefully read the comments and suggestions, and have made detail corresponding revisions. Point-by-point responses are attached below.
Point 1: Lines 22-24: If you are going to compare behaviours of juvenile and adult bats then you need to describe the adult behaviours in the Methods. And if you are directly comparing the infant and adult behaviours then you need to explain what data are being used to make these comparisons, either descriptively or empirically using statistical tests.
Response 1: Describing the behaviours of adult bats is just a way to better understand the behaviours of juvenile bats to show that there is a difference between juvenile and adult bats. The statements in the article have been changed. Please see lines 24-25.
Point 2: Lines 37-38: Include citation number immediately after the author’s name:
“Tinbergen [1] established behavioural development as one of the four main problems in behavioural biology.”
Response 2: The citation number has been added to the author's name. Please see lines 38-39.
Point 3: Lines 53-55: I would suggest using the term ‘nocturnal invertebrates’ rather than “night-flying insects”. Some bat species glean non-flying prey from vegetation, including taxa that are not insects, e.g. spiders.
Response 3: This is a great suggestion. The statements in the article have been changed. Please see lines 55-56.
Point 4: Lines 44-71: The second paragraph is very long. I would suggest splitting it into to shorter paragraphs. A logical place may be line 57:
“There are many in-depth studies on the flight development…”
Response 4: This is a very useful advice. The statements in the article have been changed.
Please see lines 45-73
Point 5: Lines 59-61: This sentence probably need a reference.
Response 5: Thank you very much for your advice. The statements in the article have been changed. Please see lines 63-64.
Point 6: Lines 72-73: This statement needs to be referenced.
Response 6: Thank you very much for your advice. The statements in the article have been changed. Please see lines 74-75.
Point 7: Line 84: Looks like there’s been an autocorrect of the spp name here:
“Vespertilio sine12nsis”
Response 7: I'm sorry about that. It was a mistake. It was an error in the writing process. The statements in the article have been changed. Please see line 86.
Point 8: Lines 89-91: Would be useful to include the mean adult body mass of the spp here.
Response 8: This is a very useful advice. Values of mean ± standard deviation of adult body weight have been added to the article. Please see lines 91-92.
Point 9: Lines 96-99: In this sentence I would add colon before the first prediction, and then a comma before the second prediction.
“We predicted that: 1) the postnatal development of the behaviours of Asian parti-coloured bat pups was related to their physical condition and age, and 2) the behaviours of the pups before flight affected the acquisition of flight.”
Response 9: Thank you very much for your advice. The statements in the article have been changed. Please see lines 99-101.
Point 10: Line 102: You don’t need to include the spp. name again here, as you have already provided it in line 84.
Response 10: Thank you very much for your advice. The statements in the article have been changed. Please see line 104.
Point 11: Lines 102-104: Do you have a study you can cite here? If not, probably worth including a reference to the author’s ‘unpublished data’.
Response 11: My pre-experimental field observations are included here and appropriate references have been added here. Please see line 106.
Point 12: Line 105: spell ‘six’ rather use the numeral (6).
Response 12: Thank you very much for your advice. The statements in the article have been changed. Please see lines 107-119.
Point 13: Lines 107-108: Was the laboratory an indoor or outdoor facility?
Response 13: The laboratory mentioned in the article is an indoor laboratory. The statements in the article have been changed. Please see lines 109-110.
Point 14: Lines 115-117: So were there three female bats in Group 1, and three bats in Group 2? Do adult female Asian parti-coloured bats normally give birth to single young, or twins? Presumably they have twins, given there were 11 pups from 6 mothers? Do did one pup die, or did one of the mothers only have a single pup?
Response 14: The first and second groups of females had the same number of three bats. Adult female Asian parti-coloured bats normally give birth to or twins. No young bats died in the experiment and one of the mothers in the experiment had only one pup.
Point 15: Lines 120-121: List the license/permit number that the work was conducted under here. What about approval from the relevant Animal Ethics Committee? Presumably both Northeast Normal University and Jilin Agricultural University require any research involving the capture and handling of wildlife animals to be approved by their Animal Ethics Committee?
Response 14: I really appreciate the advice. Permission documents related to the experiment have been sent to the editor of the journal.
Point 16: Lines 133-135: I’d recommend including the initials of the two observers here in brackets.
Response 16: It's a very useful suggestion. The statements in the article have been changed. Please see line 137.
Point 17: Line 135-136: Were the ring markers individually colour-coded? I presume they would have too small for the observers to be able to read any sort of number/ID code?
Response 17: It's a very useful suggestion. We rely on the position and number of ring markers to identify individuals. For example, left one, right two. It is indeed difficult to identify an individual in a video by colour or ID code. The statements in the article have been changed. Please see lines 140-141.
Point 18: Lines 164-166: Can you clarify whether this observation has been made during previous studies, or by the authors during this study.
Response 18: This statement is intended to explain why we have chosen to explore the relationship between flapping wings and spreading wings behaviours of young bats and their flight ability. In this experiment, as well as in previous pre-experiments, we have observed that the only wing-related behaviours of young bats are flapping wings and spreading wings behaviours. Therefore, we think that this may be related to adult flight behaviour, as both are wing-motion and similar in their behaviour patterns.
Point 19: Lines 168-169: Only one study is referenced, so I’d say that “A previous study…”, instead of “Previous studies…”.
Response 19: Thank you very much for the suggestion. The statements in the article have been changed. Please see line 174.
Point 20: Line 186: The citation at the end of the caption (Edward 2019) is formatted differently to the journal style guide.
Response 20: Thank you very much for the suggestion. The statements in the article have been changed. Please see line 192.
Point 21: Lines 306-310: In the captions for figures 5, 6 & 7 insert a bracket after A), and then B).
Response 21: Thank you very much for the suggestion. The statements in the article have been changed. Please see lines 309-344.
Point 22: Line 296: This text is in the Results, so I don’t think you need to start the sentence with “The results showed that…”
Better to start this sentence with: “The number of occurrences of…”
Response 22: Thank you very much for the suggestion. The statements in the article have been changed. Please see line 297.
Point 23: Line 321: Again, just start this sentence with: “The correclation analysis showed that…”
Response 23: Thank you very much for the suggestion. The statements in the article have been changed. Please see line 332.
Point 24: Lines 323-324: Swap “try” with ‘tried to fly earlier’.
Response 24: Thank you very much for the suggestion. The statements in the article have been changed. Please see line 335.
Point 25: A general comment – the first two paragraphs in the Discussion are very long. Id recommend breaking these up into shorter paragraphs.
Response 25: Thank you very much for the suggestion. These two paragraphs have been broken up into shorter paragraphs. Please see lines 348-405.
Point 26: Lines 338-340: When you say adult bats, are you referring to the frequency of behaviours recorded when the infant bats were older, i.e. later in the experiment (i.e. behaviours of infant bats recorded on day number 25)? I can t se anywhere in the Methods or Results any mention of behaviours of the 11 infant bats being compared to the behaviours of the 6 adult mothers being recorded. None of your models compared behaviours of infant bats with behaviours of their adult mothers. If you re going to formally compare infant and adult behaviours then this needs to be described in the Methods and Results. If you re just talking about anecdotal observations that s fine, but you need to explicit about this, and then you can t say that infant behaviours were significantly different to adult behaviours.
Response 26: It's a very useful suggestion. The adult bats mentioned here are mothers of young bats. The observations are only discussed here to show that the behaviours of young bats are different from those of adult bats. The statements in the article have been changed. Please see lines 350-351.
Point 27: Lines 343-344: Did you use an ultrasonic detector to check whether they emitted an ultrasonic vocalisations? If not, then need to say here that the infant bats did not make any sounds that were audible to the authors.
Response 27: I'm sorry about that. It was an error in the writing process. In previous pre-experiments we used an ultrasonic recorder. The results of the pre-experiment showed that the sounds made by young bats were not well collected, so the ultrasonic recorder was not used in this experiment. The statements in the article have been changed. Please see lines 357-359.
Point 28: Lines 343-345: Again, did you use an ultrasonic detector to record ultrasonic volalisations made by adult bats? Or is this a hypothesized behaviour?
Response 28: This is a very critical suggestion. The statements in the article have been changed. Please see lines 357-359.
Point 29: Lines 349-351: Again, here you’re talking about observations of adult bat behaviours, but you have not described anything about these observation s in the Methods or Results sections. Even if these observations were essentially anecdotal, if you’re going to use them in the Discussion to directly compare with the infant bevaviours that you describes and analysed then you need to also describe the adult bahviours in the Methods, and at least provide some sort of descriptive summary of the adult behaviours that you observed in the Results.
Response 29: This is a very critical suggestion. The entire experiment was focused on the behavioural development of young bats. The description of the behaviours of adult bats is intended to show that these behaviours are unique to young bats. Even though adult bats exhibit similar behaviours, the extent of the behaviours and the context in which they occur are different from those of juvenile bats. These statements in the article may be inaccurate and ambiguous. I've changed these statements in the article. Please see lines 363-365.
Point 30: Lines 357 & 359: Include the citation number each time you refer to Oppenhiem [41].
Response 30: Thank you very much for the suggestion. The statements in the article have been changed. Please see lines 371-373.
Point 31: Line 408: I’d be good to include citations to a couple of example studies that have used the Powers [35] method at the end of this sentence.
Response 31: Thank you very much for the suggestion. The statements in the article have been changed. Please see lines 426-427.
Point 32: Lines 408-416: I found this paragraph a bit difficult to follow. I’d suggest a rewrite to make it a bit clearer.
Response 32: It's a very useful suggestion. The differences between the two experimental methods listed here are intended to illustrate that the method chosen in the experiment is closer to the development of young bats in their natural state. These statements in the article may be inaccurate and ambiguous. I've changed these statements in the article. Please see lines 426-435.
Point 33: Lie 248: Include the citation number when referring to Burghardt [51].
Response 33: Thank you very much for the suggestion. The statements in the article have been changed. Please see lines 449-451.
Point 34: Lines 435-437: I think you probably need to add here that you are proposing this, i.e.:
Although the body is not yet functionally developed enough for flight, we propose that wing flapping and spreading behaviours during development, similar to play behaviour, may prepare young bats for flight.”
Response 34: It's a very useful suggestion. The statements in the article have been changed. Please see lines 457-459.
Point 35: Lines 439-440: But you didn’t describe any adult behaviours in the Methods or Results, and didn’t compare the bevaviours of infants with those of adults. Also, I’m not sure that the term “serious behaviour” is appropriate? I can’t see any way that you could objectively quantify whether any specific behavior of an infant or adult bat was “serious” or not.
Response 35: This is a very critical suggestion. I'm sorry that questions like this about adult bat behaviours come up so many times in the article. Serious behaviour is contrasted with play behaviour and refers to behaviour that has a clear purpose. I realize that statements like this in the article can be confusing and misleading, and I've made changes to the article in the appropriate places. Please see line 464.
Point 36: Lines 443-445: delete “, in order to protect bat populations” from the end of this sentence, it’s not clear how this relates?
Response 36: Thank you very much for the suggestion. The statements in the article have been changed. Please see line 469.
Point 37: Lines 445-450: This is a very long sentence that I would suggest might be better broken into two sentences.
Response 37: Thank you very much for the suggestion. The statements in the article have been changed. Please see lines 470-473.
Point 38: Line 459: Better to say that “Kangkang Zhang and Heng Liu analysed the data”
Response 38: Thank you very much for the suggestion. The statements in the article have been changed. Please see line 487.
Point 39: Lines 451-452: These videos are very helpful in providing the reader with a visual depiction of the four different behaviuours. Reference should be made to these four videos in the Methods section when you first describe the four different behaviours – I think you want the reader to watch the videos before they go on to read the Results and Discussion sections.
Response 39: This is a very helpful suggestion and I've added instructions to the Methods section of the article. Please see lines 140-141.
Point 40: Be consistent with the dash size used in page numbers. Some references use 1-10, others 1–10.
Response 40: Thank you very much for the suggestion. The statements in the article have been changed.
Point 41: Line 492: Reference 12 – Jones is spelt inccorectly (Jonesg).
Response 41: Thank you very much for the suggestion. The statements in the article have been changed. Please see line 522.
Point 42: Line 497: Reference 14 – the year should be in bold.
Response 42: Thank you very much for the suggestion. I checked the magazine's requirements, which state that the year of the book citation is not to be bolded.
Point 43: Line 527: Reference 26. Unclear whether this is a book, a book chapter, or a journal paper?
Response 43: Thank you very much for the suggestion. It’s a journal paper.
Point 44: References 31 & 33: Why are the titles of these two journals abbreviated, but other journal titles are presented in full?
Response 44: Thank you very much for the suggestion. I added the full title of the journal.
Point 45: Line 570: Reference 41 – All words in the Journal title should be chaptalized.
Response 45: Thank you very much for the suggestion. The statements in the article have been changed. Please see line 600.
Point 46: Lines 572-573: Reference 42 – What journal is this paper from?
Response 46: Thank you very much for the suggestion. I added the title of the journal. Please see line 602.
Point 47: Line 586: Reference 48 – Delete the full stop between the end of the article title and the journal name.
Response 47: Thank you very much for the suggestion. The statements in the article have been changed. Please see line 617.